# Correlation Between Antihypertensive Drugs and Survival Among Patients with Pancreatic Ductal Adenocarcinoma

**DOI:** 10.3390/cancers16233945

**Published:** 2024-11-25

**Authors:** Natalia Kluz, Leszek Kraj, Paulina Chmiel, Adam M. Przybyłkowski, Lucjan Wyrwicz, Rafał Stec, Łukasz Szymański

**Affiliations:** 1Department of Gastroenterology and Internal Medicine, Medical University of Warsaw, 02-091 Warsaw, Poland; nataliakluz99@gmail.com (N.K.); adam.przybylkowski@wum.edu.pl (A.M.P.); 2Department of Oncology, Medical University of Warsaw, 02-091 Warsaw, Poland; rafal.stec@wum.edu.pl; 3Department of Molecular Biology, Institute of Genetics and Animal Biotechnology, Polish Academy of Sciences, 05-552 Garbatka, Poland; gp.chmiel@gmail.com (P.C.); l.szymanski@igbzpan.pl (Ł.S.); 4Department of Oncology and Radiotherapy, Maria Sklodowska-Curie National Cancer Research Institute, 02-781 Warsaw, Poland; lucjan.wyrwicz@pib-nio.pl

**Keywords:** pancreatic cancer, angiotensin-converting enzyme inhibitors, ACEi, angiotensin II receptor blockers, ARBs, calcium channel blockers, CCBs, beta-blockers, BBs

## Abstract

Despite notable advancements in systemic treatment for oncology in recent years, pancreatic ductal adenocarcinoma (PDAC) continues to demonstrate resistance to the majority of available systemic therapies. The median overall survival rate for this diagnosis is between 10 and 12 months. Surgery remains the sole avenue for achieving a definitive cure. In the absence of optimal systemic options, repurposing existing antihypertensive drugs may prove to be a beneficial strategy.

## 1. Introduction

The diagnosing and treatment of pancreatic ductal adenocarcinoma (PDAC) presents significant challenges due to its aggressive nature, early tendency to metastasize, and resistance to therapies. PDAC is renowned for its rapid infiltration of adjacent tissues and rapid dissemination, which contributes to its high mortality rate [1]. Several major guidelines recommend the FOLFIRINOX regimen as a first-line treatment, including ESMO 2019, ASCO 2018–2020, NICE 2018, SEMO 2020, and NCCN 2023. However, it is important to note that due to its higher toxicity profile, this regimen is the most suitable for otherwise healthy patients (ECOG 0-1) [2,3,4,5]. Gemcitabine monotherapy is advised for ECOG 2 patients or those with significant comorbidities, such as ischemic heart disease, COPD, or poorly controlled hypertension, when risks associated with FOLFIRINOX outweigh the potential benefits [6,7,8,9]. For patients with metastatic PDAC who cannot tolerate oxaliplatin and irinotecan, gemcitabine combined with capecitabine may be suggested, though its impact on survival is modest [2].

Furthermore, the tumor’s microenvironment, typified by desmoplasia and robust immunosuppression, fosters the expeditious proliferation of cancer cell clones, rendering them susceptible to developing resistance to systemic therapies and radiotherapy with remarkable swiftness [10]. The substantial molecular and cellular heterogeneity of pancreatic cancer further complicates its treatment [11]. The incidence of PDAC is rising, marked by swift disease progression and poor survival outcomes. In regard to healthcare strategies, it is of paramount importance to acknowledge that approximately one-third of the PDAC burden is attributable to risk factors like smoking, alcohol consumption, obesity, physical inactivity, and diet [12,13]. These factors are known to elevate cancer risk by inducing oxidative stress and inflammation, which can contribute to tumorigenesis [14].

Two studies have investigated the correlation between diabetes mellitus or insulin resistance and the age of PDAC onset [15,16]. The Italian prospective study found no significant difference in the mean age of PDAC onset between patients with and without insulin resistance (67.1 years vs. 66.8 years; *p* = 0.8; N = 296). However, patients with insulin resistance were more likely to have a cancer stage of ≥3 (23.2% vs. 14.2%, *p* = 0.053) and residual disease after surgery (R1 56.4% vs. 38%, *p* = 0.007) [15]. Another study comparing the risk of PDAC across different age groups found no significant interaction between diabetes and the age at which PDAC develops (<45 years vs. <60 years; *p* = 0.84; N = 1954). Nevertheless, diabetes mellitus did increase the risk of PDAC before age 60, though not before age 45 [16]. Metabolic syndrome (MS), which includes visceral obesity, hyperglycemia, atherogenic dyslipidemia, and hypertension, is associated with elevated risks of both cardiovascular diseases and PDAC [17]. Several studies have underscored this fact [18]. For example, a study of the UK Biobank cohort found that increased abdominal waist circumference and higher blood glucose levels, key components of MS, are independently associated with a higher risk of PDAC [19]. Additionally, a study using the Korea National Health Information Database found a significant rise in pancreatic cancer risk (HR = 1.31; 95% CI, 1.09–1.56; *p* < 0.001) among individuals with four or more MS components [20]. A large nationwide cohort study by Park et al., involving more than eight million participants, showed that persistent MS significantly increased the risk of pancreatic cancer [21]. Interestingly, the risk was higher in patients who had recovered from MS compared to those who had never been diagnosed with MS, though it remained lower than in those with persistent MS.

Considering the extensive range of medications currently available for managing MS, repurposing these drugs could prove to be a relevant avenue for exploring concomitant treatments in PDAC [22]. Given the unfavorable prognosis of PDAC and the limited efficacy of novel anticancer strategies, such as targeted therapies and immune checkpoint inhibitors, it is imperative to pursue alternative avenues of investigation. Drug repositioning, which involves identifying novel clinical applications for existing drugs originally approved for non-oncological indications, represents a promising strategy. Antihypertensive drugs may offer promise as a means of influencing cancer development, either directly or indirectly. In vitro studies have demonstrated that antihypertensive drugs can augment a cytotoxic effect against chemoresistant cell lines, inhibit cell growth, and enhance chemosensitivity in a range of cancer types [23,24,25,26]. Furthermore, these drugs are well tolerated, orally administered, and no longer patented, which makes them more affordable than many other cancer treatments [27].

This review aimed to examine the potential of repurposing antihypertensive drugs as an adjuvant therapeutic option in PDAC and evaluate whether medications used to treat comorbidities can enhance the efficacy of chemotherapy.

## 2. Resistance Mechanisms in PDAC: Key Pathways in Chemotherapy Resistance

### 2.1. Molecular/Genetic Patterns of Chemoresistance

The COMPASS trial served as the foundation for the initial genetic profiling of this tumor, as it evaluated genetic markers that predicted response to chemotherapy in patients with advanced pancreatic cancer. The results demonstrate that chemotherapy response rates vary according to transcriptomic profiles [28]. In PDAC, mutations in *KRAS* are the most commonly observed oncogenes (also defined as “driver mutations”). These mutations impair KRAS intrinsic GTPase activity, which prevents GTPase-activating proteins from converting the active, GTP-bound form of KRAS into its inactive, GDP-bound state. Clinical studies have indicated that *KRAS* mutations can be a marker of poor prognosis in PDAC [29]. In advanced PDAC, *KRAS* mutations upregulate the expression of the antioxidant factor *NRF2* in response to elevated oxidative stress levels within PDAC cells, contributing to chemoresistance [30].

Other genes, such as *CDKN2A*, *SMAD4*, and *TP53*, also play crucial roles in the tumorigenesis and metastasis of pancreatic cancer [31]. Mutations in *CDKN2A*, found in about 46–60% of cases, lead to cell cycle dysregulation [31,32]. Furthermore, *SMAD4* mutations, occurring in 31–38% of cases, typically emerge in later stages and contribute to cancer progression by diminishing the inhibitory effects of TGF-β signaling [31,33]. Approximately 50–74% of pancreatic cancers harbor inactivating mutations in *TP53*, which disrupts DNA damage response and cell cycle regulation, allowing cells to evade apoptosis [31]. *TP53* mutations consist of about two-thirds missense mutations and one-third truncating mutations, which affect mRNA degradation mechanisms [34]. These mutations, particularly gain-of-function variants, alter the tumor microenvironment, promoting tumor proliferation and resistance to chemotherapy [34,35].

Capello et al. investigated CES2, an enzyme that activates irinotecan into SN-38, through in vitro and in vivo models, alongside comprehensive analyses of genetic databases, proteomics, and tissue microarrays. They found that high *CES2* expression was linked to longer overall survival (OS) and progression-free survival (PFS) in patients with resectable and borderline-resectable pancreatic cancer treated with FOLFIRINOX in the neoadjuvant setting [36]. Notably, this study is the first to report an association between the molecular characteristics of pancreatic tumors and the outcomes of FOLFIRINOX treatment. However, the univariate and multivariate analyses were limited by the small sample size (n = 22).

Several signaling pathways, including RAS, PI3K/AKT, NF-κB, JAK/STAT, Hippo/YAP, and Wnt, play roles in cancer-related processes such as cell proliferation, differentiation, and migration. These pathways have also been linked to PDAC development, prognosis, and treatment resistance [37,38,39,40].

*RAS* activation can drive signaling pathways involved in cell transformation, proliferation, metastasis, and inflammatory signaling via NF-κB and STAT3 [41,42]. Recent studies have shown that *KRAS* mutations play a crucial role in the metabolic reprogramming of cancer cells, steering them toward an anabolic metabolism essential for biomass production and supporting unconstrained proliferation [43]. In KRAS-driven cancers like PDAC, calmodulin plays a dual role in regulating the activation of the MEK/ERK and PI3K/AKT pathways. Phosphorylated calmodulin can enhance KRAS-induced activation of the PI3K/AKT signaling pathway by strengthening the interaction between *KRAS4b* and *PI3Kα* [44]. The PI3K/AKT survival pathway has also been implicated in gemcitabine resistance in conjunction with *ILK* [45,46]. *ILK* increases gemcitabine chemoresistance in PDAC cells due to a chemoprotective effect occurring in association with suppression of activity [46]. In PDAC without *KRAS* mutations, RAS activation can occur through signaling via receptor tyrosine kinases (RTKs) such as EGFR and occasionally through activation of the proto-oncogene B-Raf in a subset of patients [47]. EGFR overexpression in pancreatic cancer has been associated with gemcitabine resistance due to the transactivation of anti-apoptotic PI3K/AKT signaling [48]. Although preclinical studies showed potential, the addition of the EGFR inhibitor erlotinib to gemcitabine monotherapy increased median overall survival by less than two weeks in a phase III trial [49]. Therefore, these mechanisms highlight the complexity and robustness of RAS-driven signaling pathways in promoting PDAC’s aggressive behavior and resistance to chemotherapy. It underscores the significant challenge in effectively treating *KRAS*-mutant PDAC and suggests that targeting these pathways may be essential to overcoming resistance and improving therapeutic outcomes, despite the limited success observed with EGFR inhibitors in clinical settings.

Wnt signaling and the nuclear activity of β-catenin play significant roles in driving drug resistance in PDAC. These pathways are undoubtedly important for the proliferation, differentiation, and maintenance of CSCs [50]. CSCs are known to survive after treatment and contribute to cancer relapse and therapeutic resistance by increasing the expression of ATP-binding cassette transporters, evading apoptosis, and regulating EMT. In fact, pancreatic cancer cells that survive high doses of gemcitabine have been shown to express higher levels of CSC-related genes and exhibit EMT characteristics [50,51]. The chemoresistance of CSCs is driven by mechanisms such as the metabolic inactivation of chemotherapeutic drugs, increased drug efflux from the cells, and mutations or deregulation of drug targets. Specifically, altered drug transport activity—including overexpression of aldehyde dehydrogenase and proteasome, along with decreased expression of human ENTs and CNTs—plays a critical role in the resistance of pancreatic cancer stem cells to chemotherapy [52].

Calcium channels play an important role among the mechanisms of chemotherapy resistance in PDAC. Store-operated calcium channels (SOCs) are among the primary calcium-entry pathways in non-excitable cells. SOCs, particularly their key components *ORAI1* and *STIM1* [53], are involved in various physiological processes, including cell proliferation, development, contraction, and motility [54,55,56]. Recent studies have also highlighted the significant role of ORAI1 and STIM1 in carcinogenesis. Elevated *STIM1* expression can drive tumor progression-related processes, such as metastasis, through the abnormal activation of Ca^2+^-dependent signaling pathways, and it is associated with reduced patient survival across multiple cancer types [57]. In pancreatic cancer, *STIM1* and *ORAI1* inhibit apoptosis [58]. Therefore, blocking *STIM1* and *ORAI1* has been shown to enhance apoptosis in pancreatic cancer cells induced by chemotherapeutic agents like 5-fluorouracil (5-FU) and gemcitabine [59].

### 2.2. EMT-Mediated Resistance

EMT is a process in which cancer cells acquire a mesenchymal phenotype, increasing their invasiveness and resistance to therapy. EMT is marked by both morphological and genomic alterations, with key mesenchymal transcription factors such as Snail, Slug, and Zeb1 playing crucial roles. For example, inhibiting Zeb1 can reduce TGF-β signaling and associated metabolic changes, thereby impacting cancer cell colonization [60]. There is a significant negative correlation between E-cadherin and Zeb-1, which is closely associated with chemoresistance. Zeb-1, along with other EMT modulators, plays a key role in sustaining drug resistance in human pancreatic cancer cells. Silencing the Zeb-1 gene leads to an upregulation of E-cadherin protein expression, which in turn restores drug sensitivity by enhancing the expression of epithelial markers such as EVA1 and MAL2 [61]. Additionally, the knockdown of Slug has been shown to enhance sensitivity to gemcitabine in pancreatic cancer cells [62].

EMT is a significant contributor to chemoresistance in PDAC, as it endows cancer cells with enhanced survival mechanisms, including increased efflux of chemotherapeutic drugs, resistance to drug-induced apoptosis, and an active DNA repair capacity. Recent evidence suggests that EMT is tightly associated with the acquisition of a CSC-like phenotype, which is inherently resistant to conventional chemotherapy. For instance, gemcitabine-resistant PDAC cells often exhibit elevated levels of EMT transcription factors and a reduction in epithelial markers such as E-cadherin. This shift towards a mesenchymal phenotype is linked to increased treatment resistance and tumor recurrence. Tumor microenvironments rich in EMT mediators, including TGF-β1, EGF receptor, and hypoxia, present further challenges for treatment.

The combined effects of survival pressures, extracellular matrix interactions, and paracrine signals from supportive cells significantly drive cancer-associated EMT [63,64]. Moreover, growth factor signaling from activated fibroblasts, particularly through pathways like Notch, Wnt/β-catenin, and Hedgehog, plays a crucial role in sustaining the EMT process and promoting tumor growth and metastasis [64]. Due to lack of CD8+ T cells in the tumor microenvironment of PDAC, the strategies to inhibit the immune checkpoints fail to demonstrate efficacy in advanced PDAC patients [64].

Inflammatory markers amplify these effects, while myeloid-derived suppressor cells (MDSCs), regulatory T cells (Tregs), and M2 macrophages simultaneously suppress effector T cells, dendritic cells, and CD8+ cytotoxic T cells, hindering their ability to target and destroy tumor cells.

Additionally, EMT mediators have been found to alter drug absorption profiles by regulating the expression of drug influx and efflux transporters, further complicating treatment strategies. For example, the loss of epithelial markers and the upregulation of mesenchymal markers can lead to reduced expression of drug transporters like ENT1, which is crucial for gemcitabine uptake. This alteration contributes significantly to the chemoresistance observed in PDAC. Strategies aimed at targeting the desmoplastic stroma have shown promise, with an increasing focus on reducing pressure, enhancing vascularization, and improving the diffusive capacity of ECM [65,66]. Targeting EMT-related pathways and mediators offers a potential therapeutic approach to overcoming drug resistance in PDAC, paving the way for more effective treatment options.

### 2.3. Tumor Microenvironment

There is growing interest in targeting the tumor microenvironment (TME) due to its role in resistance to systemic therapies and its potential to enable more effective treatments. Pancreatic ductal adenocarcinoma (PDAC) is often characterized as an “immunologically cold” tumor due to its deficiency in the high mutation burden needed for the immune system to recognize tumor cells through tumor-specific antigens [67]. This low immunogenicity results in inadequate cancer antigen presentation, leading to a weak or absent immune response and insufficient T cell infiltration [68].

Moreover, PDAC alters the surrounding tissues and immune cells, creating a dense, fibroblastic environment that further suppresses anti-tumor immunity. The TME in PDAC consists of cancerous pancreatic cells, supportive stromal cells, and a dense extracellular matrix (ECM), forming a physical barrier that restricts drug delivery to the tumor cells. This barrier, combined with poor vascularization, results in reduced perfusion and drug distribution, significantly limiting the efficacy of chemotherapeutic agents. Additionally, the TME supports immune evasion by excluding immune cells and fosters an environment that is resistant to treatment, adaptable to therapies, and conducive to cancer spread.

A hallmark of PDAC is desmoplasia, characterized by the presence of CAFs derived from PSCs, endothelial cells, and inflammatory cells [69]. CAFs are known to secrete ECM proteins and cytokines that promote survival signaling pathways, like NFκB and STAT3, and enhance the EMT, further aiding in the evasion of apoptosis and perpetuating resistance to chemotherapy [70]. Moreover, PSCs contribute to the formation of a dense stromal matrix that can both inhibit effective drug delivery and support cancer cell survival through paracrine signaling pathways, including Hedgehog and SDF-1α/CXCR4 signaling, which are linked to chemoresistance [71,72]. The low microvascular density not only restricts nutrient and oxygen supply but also reduces drug delivery, allowing cancer cells to survive under harsh conditions and resist cytotoxic therapies [73].

In contrast to normal differentiated cells, which primarily utilize mitochondrial oxidative phosphorylation to meet their energy demands, most cancer cells predominantly rely on aerobic glycolysis, a phenomenon known as “the Warburg effect” [74]. This shift towards glycolysis also plays a crucial role in facilitating interactions within the tumor stroma. Specifically, lactate may serve as a key mediator in tumor–stroma communication and symbiotic energy exchange between different cell compartments within the tumor [75]. Hypoxia directly enhances lactate production and excretion, driven by changes in mitochondrial redox status due to reduced oxygen availability [76]. Hypoxic cancer cells produce lactate, which is exported to the extracellular space via the MCT-4 transporter and subsequently taken up by normoxic cancer cells through MCT-1, where it is used for oxidative metabolism, thereby conserving glucose for hypoxic cells [77]. The hypoxic tumor microenvironment, which activates HIF-1α, induces similar aberrant signaling pathways as oncogene activation of *KRAS* and *MYC*, as well as inactivation of the tumor suppressor gene *TP53* [78]. Furthermore, an acidic TME may contribute to resistance to gemcitabine by promoting EMT [79]. Furthermore, the hypovascularity characteristic of the PDAC microenvironment limits the effective delivery of chemotherapeutic agents, exacerbating treatment resistance. This interplay between hypoxia, stromal barriers, and immune exclusion makes the TME a formidable obstacle in the treatment of PDAC, underscoring the need for strategies that target both the tumor and its microenvironment to improve therapeutic outcomes.

## 3. Repurposing Drugs for the Treatment of Pancreatic Cancer

Antihypertensive drugs are broadly categorized into groups based on their mechanisms of action: those targeting the renin–angiotensin system (RAS) by inhibiting angiotensin-converting enzyme (ACE), blocking the angiotensin type 1 receptor (AT1R), directly inhibiting renin, or antagonizing aldosterone binding to its receptor; and those that block calcium channels and β-adrenoceptor antagonists.

In the following discussion, we will focus on three principal groups of antihypertensive drugs used as adjuvants in cancer treatment, based on in vitro, in vivo, and clinical evidence [80]. The cellular mechanisms by which these antihypertensives exert their effects on cancer cells are depicted in Figure 1, and the role of mentioned drugs in PDAC treatment are summarized in Table 1.

### 3.1. Renin Pathway Inhibitors—ACEIs and ARBs

The use of angiotensin-converting enzyme inhibitors (ACEIs) and angiotensin receptor blockers (ARBs) has been associated with improved outcomes in patients with various types of cancer, including non-small cell lung cancer, pancreatic cancer, and breast cancer [80]. This association is supported by a growing body of preclinical studies that highlight the role of RAS signaling in the development, growth, and progression of cancer [99]. These findings have spurred further investigations into the effects of RAS inhibitors (RASi) in cancer patients, both in retrospective analyses and prospective trials. Research on human PDAC specimens has shown that angiotensin type 1 receptor (AT1R) expression is significantly elevated compared to normal pancreatic tissue, while the expression of angiotensin type 2 receptor (AT2R) is slightly reduced in neoplastic ductal epithelium [70]. The study also demonstrated that treatment with a selective AT2R agonist in murine PDAC grafts reduces tumor growth by inducing apoptosis in PDAC cells [70]. This suggests a potential benefit of using AT2R agonists in combination with AT1R antagonists for PDAC treatment.

Blocking AT1R has been shown to reduce obesity-induced fibrosis and tumor progression and improve response to chemotherapy. AT1R blockade also decreases the number of TANs and Tregs while increasing the number of CD8+ T cells by inhibiting the activation of PSCs and subsequently reducing IL-1β expression [81]. Additionally, in an orthotopic model of pancreatic cancer, losartan-mediated inhibition of aberrant TGF-β activity resulted in decreased collagen deposition and Treg accumulation [82].

RASi therapy has shown clinical benefits in various cancer types, ranging from slowly progressing prostate cancer to aggressive forms like glioblastoma and pancreatic cancer. Since 2010, RAS blockade has been observed to positively influence pancreatic cancer mortality, leading to a phase I clinical trial assessing the combination of candesartan and gemcitabine, which was found safe for further phase II trials [83,84]. Several clinical trials have explored the role of ARBs in combination with chemotherapy for PDAC.

In a phase II trial involving 35 patients with advanced pancreatic cancer, those receiving 16 mg of cadesartan experienced a modest but significant increase in progression-free survival compared to those taking 8 mg (4.6 vs. 3.5 months) [85]. In another single-arm phase II trial, losartan combined with a chemotherapeutic cocktail (FOLFIRINOX) and followed by chemoradiotherapy (CRT) helped achieve complete surgical resection in patients with locally advanced PDAC [86]. Preliminary results from this trial show that R0 resection was achieved in 13 out of 25 patients (52%), representing a significant improvement over previously reported R0 resection rates of 23–24% with neoadjuvant FOLFIRINOX (FFX) and CRT for locally advanced PDAC [100,101]. The median OS was 33 months, with a 2-year survival rate of 65% for all patients and 83% for those who underwent resection [86]. Although this was a single-arm study, a parallel phase II trial comparing FOLFIRINOX without losartan, followed by chemoradiotherapy, reported a similar rate of complete resection in patients with borderline resectable pancreatic adenocarcinoma [102]. Boucher et al. performed a gene expression and immunofluorescence (IF) analysis where they used archived surgical samples from patients treated with LOS+FFX+CRT and patients treated with FFX+CRT. Incorporating LOS into FFX+CRT reduced genes associated with pro-invasion and immunosuppression, which correlated with enhanced OS in PDAC patients. Lesions from responders treated with the LOS+FFX+CRT combination showed a reduction in Tregs, decreased C-FOXP3 levels, and an increase in CD8+ T cells [87]. This suggests that losartan may enhance resectability.

Multiple studies have demonstrated that RAS inhibitors can successfully normalize the fibrotic stroma. Co-injection of cancer cells with stromal cells results in increased tumor size and fibrosis, while treatment with ARBs mitigates these effects. Losartan inhibits collagen I production by CAFs and reduces stromal collagen and HA in various desmoplastic tumor models by decreasing pro-fibrotic signaling via TGF-β, connective tissue growth factor, HA synthase 1 and 3, and endothelin-1. Consequently, losartan reduces solid stress and enhances vascular perfusion, leading to decreased tumor hypoxia and improved distribution and efficacy of anticancer drugs and nanotherapeutics [88]. Similarly, inhalation delivery of losartan and telmisartan has been shown to reduce active TGF-β and collagen I expression while increasing the intratumoral distribution of nanoparticles [103]. Additionally, the interaction between TANs, adipocytes, and PSCs promotes tumor desmoplasia and pancreatic cancer growth in obesity [81].

In a study conducted by Zhou et al., using a high-throughput screening platform of gemcitabine-resistant patient-derived organoids (PDOs), ibesartan was found to reduce resistance to chemotherapy. Both in vitro and in vivo studies demonstrated that ibesartan sensitizes PDAC tumors to chemotherapy by suppressing c-Jun expression through the inhibition of the Hippo/YAP1/TAZ signaling pathway. The study also revealed that c-Jun plays a critical role in enhancing tumoral iron metabolism and promoting cancer cell stemness in PDAC, which can be inhibited by ibesartan [89].

In patients with PDAC, lisinopril has been also found to extend overall survival time independently of chemotherapy. It has also been suggested that ACEIs may reduce the malignant potential of cancer cells and stimulate the immune microenvironment in PDAC patients [90]. Conversely, a retrospective analysis indicated that the use of renin–angiotensin system inhibitors (RASis) was associated with longer OS in pancreatic cancer patients with resected primary tumors (median OS of 36.3 months vs. 19.3 months) and locally advanced tumors (median OS of 11.3 months vs. 9.3 months), but not in patients with metastatic cancer. The data suggest that lisinopril, the most commonly used ACEI in this cohort, helped normalize the ECM, downregulate genes involved in cancer progression (such as those in the Wnt and Notch signaling pathways), and upregulate genes associated with T cell and antigen-presenting cell activity [104].

### 3.2. Calcium Channel Blockers—CCBs

Various calcium channels have been implicated in the mechanisms of chemoresistance, including those in pancreatic adenocarcinoma cells [105]. By employing single-cell RNA sequencing and high-throughput proteomic analysis, it was found that a subset of gemcitabine-resistant tumor cells exhibited increased calcium/calmodulin signaling. Inhibition of calcium-dependent calmodulin activation led to a rapid loss of the resistant phenotype in vitro, and further single-cell RNA sequencing revealed that this was linked to reduced activation of the RAS/ERK signaling pathway. Similarly, calcium chelation or reducing calcium levels in the culture media diminished ERK activation in resistant cells and restored their sensitivity to gemcitabine. Calcium channel blockers (CCBs) such as amlodipine were also effective in reducing prosurvival ERK signaling in vitro and significantly improved the response to gemcitabine in both orthotopic xenograft and transgenic PDAC models [106,107,108,109].

Principe et al. observed that amlodipine reduced ERK activation in ex vivo slice cultures of PDAC tumors and reinstated gemcitabine sensitivity in orthotopic xenografts of gemcitabine-resistant (GR) tumor cells, leading to reduced metastases and prolonged overall survival. Additionally, in a transgenic PDAC model, amlodipine significantly enhanced the antineoplastic effects of gemcitabine, markedly improving survival compared to gemcitabine alone. Notably, cell lines, animal models, and human PDAC tissues exhibited increased *CALM2* expression following prolonged gemcitabine treatment, often accumulating at the cell membrane. Pharmacologic inhibition of calcium-dependent calmodulin activation effectively reversed gemcitabine resistance in vitro by blocking prosurvival ERK and MAPK signaling [91].

There is some research identifying that other antihypertensive drugs, including nifedipine and fendiline, show promising activity in vitro against cancer cell lines. These CCBs can decrease proliferation and inhibit growth specifically in pancreatic cancer cell lines by inducing G1 arrest. Additionally, fendiline enhances intercellular adhesions, contributing to reduced cell migration and invasion [92].

1,4-Dihydropyridine (DHP) drugs are not the only CCBs showing promise in cancer treatment. Research on verapamil has demonstrated its direct effects on pancreatic cancer cells by inhibiting their proliferation and inducing differentiation. Verapamil has also exhibited inhibitory effects on human colonic tumor cells and antiproliferative properties in various tumor cell lines by preventing DNA fragmentation, including those for medulloblastoma, pineoblastoma, glioma, and neuroblastoma [93,110].

Diltiazem has also shown potential in enhancing the effects of chemotherapy. In the context of pancreatic cancer, CCBs have been found to inhibit P-glycoprotein (also known as multi-drug resistance protein 1), an efflux pump that removes chemotherapy from cells. The antihypertensive drug has reduced its expression and increased the intracellular concentration of chemotherapeutic agents. This results in reduced metastasis and improved survival when CCBs are administered alongside chemotherapy in animal models [94].

Despite some recent in vitro studies, the use of CCBs has not yet been fully integrated into clinical oncology. However, there are some convincing clinical data to support these findings.

Previous small retrospective studies have suggested a potential link between CCB prescriptions and improved survival in pancreatic cancer. For instance, UK single-center retrospective studies found that pancreatic cancer patients concurrently taking CCBs showed improved OS [95]. A larger, retrospective cohort study using data from the Polish National Health Fund identified a significant survival benefit in PDAC patients prescribed CCBs alongside gemcitabine [96]. Notably, this Polish study focused exclusively on patients receiving chemotherapy.

The recent study expanded on previous evidence by including both patients who received neoadjuvant treatment and those who did not. Among the 6223 patients in the study, 660 were prescribed CCBs. Of these, 591 received neoadjuvant chemotherapy. In this subgroup, CCBs prescription was linked to improved OS after adjusting for multiple prognostic factors (aHR = 0.715, 0.514–0.996, *p* = 0.047). This study was the first to show that the association between CCB prescription and overall survival is more pronounced in patients undergoing neoadjuvant therapy [97].

A population-based cohort study included 703,448 patients, representing 3.3 million people. This study was the first one of sufficient size to adequately assess the association between dihydropyridine CCBs and pancreatic cancer risk. Given the long-term use of dihydropyridine CCBs in patients with hypertension, this observational study provides much-needed evidence and reassurance to both physicians and patients regarding the safety of this drug class in relation to pancreatic cancer [111].

### 3.3. Beta-Blockers—BBs

Catecholamines regulate numerous cellular and tissue functions critical for the development and progression of many common cancers [112]. Adrenergic receptors are categorized into two main classes, α- and β-receptors, each with further subdivisions [113]. The β2-adrenergic receptor (β2AR), which primarily binds epinephrine and norepinephrine, is predominantly found in airway smooth muscle and cardiac tissue [113,114]. However, its presence in other tissues, including the pancreas, has been identified more recently [115].

Beta-adrenergic receptor activation is believed to promote cell proliferation through PKA pathway activation. PKA signaling drives proliferation and inhibits apoptosis via downstream effectors such as the cAMP CREB, AP-1, and NF-κB [116]. Additionally, β-adrenergic activation increases the levels of MMP-2, MMP-9, and VEGF [117,118,119]. These molecules facilitate the degradation of extracellular matrix components and cell adhesion molecules, as well as the release and activation of growth factors [120,121,122]. Thus, various complex, receptor-mediated signaling pathways are implicated in the β-adrenergic receptor-driven process of angiogenesis, tumor proliferation, invasion, migration, and metastasis in PDAC [117,119,123,124,125].

Epidemiological and retrospective data on β-blocker (BB) use in PDAC patients show mixed results. A Swedish population-based cohort study reported reduced cancer-specific mortality among PDAC patients using BBs, whereas retrospective analyses using UK and Clinical Practice Research Datalink databases provided inconsistent findings, potentially due to unadjusted cancer staging [126,127].

A meta-analysis of 319,006 patients by Na et al. in 2018 demonstrated that BB use was associated with improved OS in pancreatic cancer, as well as in ovarian cancer (HR = 0.59, 95% CI: 0.36–0.96, *p* = 0.034) and melanoma (HR = 0.81, 95% CI: 0.67–0.97, *p* = 0.026). For pancreatic cancer specifically, the meta-analysis included two studies with 16,092 patients, showing an association between BB use and prolonged OS (HR = 0.85, 95% CI: 0.75–0.97, *p* = 0.014) [128].

Mechanistically, BBs inhibit key oncogenic pathways, including NF-κB, PI3K/Akt, and PKA. For instance, combining the β2-adrenoceptor blocker (ICI 118551) with gemcitabine has shown enhanced apoptosis in pancreatic cancer cells through regulation of pro-apoptotic (Bax) and anti-apoptotic (Bcl-2) molecules downstream of NF-κB. This combination provided superior tumor suppression compared to gemcitabine alone, highlighting a potential for synergy in chemotherapy [98].

With their established safety profile, affordability, and well-characterized pharmacology, BBs represent a promising adjunct to existing chemotherapy regimens in PDAC management. Future prospective studies are essential to explore their potential synergistic effects with chemotherapy, such as gemcitabine, and to better define the comparative benefits of selective and nonselective BBs in this setting.

## 4. Discussion

Currently, there are no highly effective treatments for PDAC. While palliative chemotherapy offers some survival benefits, almost all patients will eventually experience disease progression, and long-term survival rates remain low. Due to the limited treatment options available after initial therapy, this review corroborates and extends prior findings that antihypertensive drugs may contribute to prolonged survival in patients with pancreatic cancer. The results underscore the potential of these medications, including renin–angiotensin system inhibitors (RASis), calcium channel blockers (CCBs), and beta-blockers (BBs), to serve as adjunctive treatments in combating this aggressive disease. The capacity to employ antihypertensive drugs as adjunctive therapies could facilitate a more comprehensive approach to treatment, encompassing the targeting of multiple pathways involved in cancer progression and resistance. The multiplicity of potential signaling pathways implicated in chemoresistant pancreatic cancer may present a significant challenge in the development of an optimal therapeutic strategy. The MAPK/ERK pathway and potential mechanisms for targeting it may offer the greatest promise. The potential role of calcium ions in resistance may be a topic for discussion in clinical trials of CCBs and adjunctive chemotherapy. Furthermore, while the current treatments for pancreatic cancer are toxic and associated with a range of severe adverse effects, antihypertensive medications are widely prescribed, well-tolerated, inexpensive drugs with minimal side effects. However, to fully realize the potential of antihypertensives in pancreatic cancer treatment, future research should focus on utilizing large, multicenter databases and prospective clinical trials to conduct more powerful and comprehensive analyses. Such studies would facilitate a more profound inquiry into the impact of diverse RASi, CCB, and BB formulations, investigating the influence of dosage variations and their potential synergistic effects when combined with distinct chemotherapy regimens. Furthermore, these analyses could assist in the identification of specific patient subgroups that may derive the greatest benefit from such treatments, thereby facilitating the development of more personalized and efficacious therapeutic strategies.

## 5. Conclusions

In conclusion, the repurposing of antihypertensive drugs for the treatment of pancreatic cancer represents a promising avenue for improving patient outcomes. By capitalizing on the favorable safety profile and cost effectiveness of these drugs in conjunction with rigorous research and clinical trials, we can pave the way for more effective and less toxic treatment options for patients afflicted with this devastating disease.

## Figures and Tables

**Figure 1 cancers-16-03945-f001:**
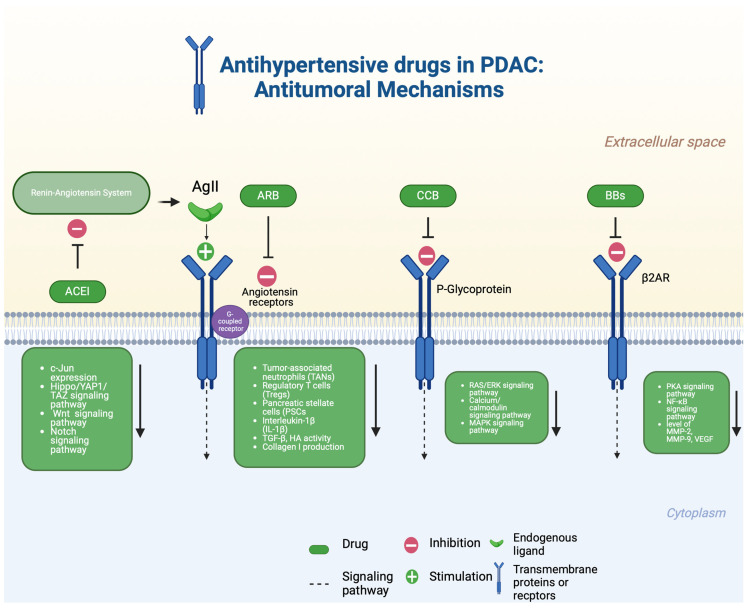
Antihypertensive drugs in PDAC: antitumoral mechanisms. In this figure, we summarize potential mechanisms through which antihypertensive drugs may aid chemotherapy through different cellular effects. Ag II, angiotensin II; HA, hyaluronic acid; ACEI, angiotensin-converting enzyme inhibitors; ARB, angiotensin I receptor blocker; CCB, calcium channel blocker; β2AR, β2-adrenergic receptor; BBs, beta-blockers. Created in BioRender.com (accessed on 20 November 2024).

**Table 1 cancers-16-03945-t001:** A summary of research on the impact of antihypertensive medication on the prognosis of pancreatic cancer. Note: 5-FU, 5-fluorouracil; RR, response rate; R0, complete resection; NK cells, natural killer cells; DC cells, dendritic cells, FFX+CRT, FOLFIRINOX followed by chemoradiotherapy; LOS+FFX+CRT, FOLFIRINOX+ losartan followed by chemoradiotherapy; GR, gemcitabine-resistant.

Research Type	Type of Therapy	Antihypertensive Drug	Antitumoral Mechanism	Outcomes	Reference
Original research (non-human study)		AT2R agonist	Stimulation of PI3K/AKT pathway increase in PDAC cell survival	-	Ishiguro S, et al., 2015 [70]
Original research (non-human study)	5-FU	Losartan	Inhibition of the protumorigenic activity of TANs and IL1β, which modulated the recruitment/activity of CD8-positive T cells and Tregs	Losartan-induced decrease in SNAIL-chemosensitizing effect	Incio J et al., 2016 [81]
Original research (non-human study)		Losartan	Abrogating aberrant TGFβ activation, decreasing microvessel density in tumors grown, reducing the activation of regulatory T cells	Inhibiting tumor growth	Arnold SA et al., 2012 [82]
Retrospective study	Gemcitabine	ACEI/ARB	-	PFS = 8.7 months in the ACEI/ARB group vs. 4.5 months in the non-ACEI/ARB group,	Nakai Y et al., 2010 [83]
Clinical trial (phase I)	Gemcitabine	Cadesartan (16 mg)	-	RR = 0%, disease control rate = 79%, respectively. PFS = 7.6 months, OS = 22.9 months	Nakai Y et al., 2012 [84]
Clinical trial (phase II)	Gemcitabine	Cadesartan (16 mg vs. 8 mg)	-	RR = 11.4%, disease control rate = 62.9%, PFS = 4.3 months, OS = 9.1 months (1-year survival rate 34,2%), PFS (16 mg) = 4.6 months vs. PFS (8 mg) = 3.5 months (*p* = 0.031)	Nakai Y et al., 2013 [85]
Clinical trial (phase II)	Neoadjuvant FOLFIRINOX+CRT	Losartan	-	R0 resection rate = 61%; PFS = 21.3 months, OS = 33.0 months	Murphy JE et al., 2019 [86]
Original research (human study)	FOLFIRINOX+CRT vs. Losartan+FOLFIRINOX+CRT	Losartan	Maturation of blood vessels and transendothelial migration of leukocytes, activation of T cells, cytolytic activity of T cells and NK cells, and DC cell activity	In comparison with FFX+CRT, LOS+FFX+CRT downregulated immunosuppression and pro-invasion genes	Boucher Y et al., 2023 [87]
Original research (non-human study)	5-FU	Losartan	Reducing stromal activity and production of matrix components and targeting all stromal components (CAFs, hyaluronan, and collagen)	Improved effectiveness of small-molecule chemotherapeutics through antimatrix effects	Chauhan VP et al., 2013 [88]
Original research (non-human study)	Gemcitabine	Irbesartan	Suppressing stemness and iron metabolism via inhibition of the Hippo/YAP1/c-Jun axis		Zhou T et al., 2023 [89]
Original research (human study)	-	Lisinopril	Reduced expression of genes involved in PDAC progression-Wnt and Notch signaling, an increased expression of genes linked with the activity of T cells and antigen-presenting cells	Longer OS independently of chemotherapy	Liu H et al., 2017 [90]
Original research (non-human study)	Gemcitabine	Amlodipine	Inhibition of prosurvival ERK and MAPK signaling—decrease in the expression of CALM2	Restoring gemcitabine sensitivity in orthotopic xenografts of GR tumor cells, reducing metastases, and extending OS enhancement of the antineoplastic effects of gemcitabine	Principe DR et al., 2022 [91]
Original research (non-human study)	-	Fendiline	Inducing G1 arrest, enhancement of intercellular adhesions, contributing to reduced cell migration and invasion	Decrease in proliferation and inhibiting tumor growth	Woods N et al., 2015 [92]
Original research (non-human study)	Gemcitabine	Verapamil	Inhibiting P-gp transporters and inducing apoptosis of stem-like SP cells in PDAC cells	Enhancement of cytotoxic effects of chemotherapeutic drugs	Zhao L et al., 2016 [93]
Original research (non-human study)	Gemcitabine/5-FU	Dilitiazem	Decrease in the expressions of stem cell markers CD24 and CD44, increase in the expressions of Bax and cleaved caspase 3, enhanced DNA fragmentation, and attenuated cyclin D1 and P-gp expressions	Enhancement of cytotoxic effects of chemotherapeutic drugs	El-Mahdy et al., 2020 [94]
Retrospective study	Gemcitabine/FOLFIRINOX	CCBs	-	Adjusted Cox regression revealed significantly improved OS—HR 0.496 (95% CI = 0.297–0.827; *p* = 0.007). Kaplan–Meier estimated median survival = 15.3 months for patients prescribed CCBs versus 10.1 months for patients not prescribed CCBs (*p* = 0.131).	Tingle SJ et al., 2020 [95]
Retrospective study	Gemcitabine	CCBs	-	A significant difference (*p* < 0.001) was observed in the median OS between patients who were prescribed CCB (n = 380; OS 9.3 months; 95% CI: 7.8–11.0) and those who were not (n = 4214; OS 7.6 months; 95% CI: 7.3–7.8), with a hazard ratio for death of 0.70 (95% CI: 0.62–0.79).	Kraj L. et al., 2017 [96]
Retrospective study	Neoadjuvant therapy: FOLFIRINOX/vgemcitabine with nab-paclitaxel/ 5-FU/gemcitabine-based regimen.	CCBs	-	Median OS = 27.5 months in those receiving neoadjuvant chemotherapy (30.7 months for those prescribed CCB, 26.5 for those not prescribed CCB)	Fong ZV et al., 2024 [97]
Original research (non-human study)	Gemcitabine	BBs	Regulation of pro-apoptotic (Bax) and anti-apoptotic (Bcl-2) molecules; downstream of NF-κB signaling pathway	Enhanced apoptosis in pancreatic cancer cells—superior tumor suppression compared to gemcitabine alone, highlighting a potential for synergy in chemotherapy	Shan T et al., 2011 [98]

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
