# Peer review of "Correlation Between Antihypertensive Drugs and Survival Among Patients with Pancreatic Ductal Adenocarcinoma"

_cancers, 2024, doi:10.3390/cancers16233945_

Round 1
Reviewer 1 Report
Comments and Suggestions for Authors
Pancreatic ductal adenocarcinoma (PDAC) demonstrates resistance to commonly-used chemotherapy. Meanwhile, metabolic syndrome (MS) such as hypertension and obesity contributes to elevated risks of both cardiovascular diseases and PDAC. Thus, this review emphasizes that the repurposing of antihypertensive drugs may prove to be beneficial chemosensitizer for the treatment of unresectable PDAC patiens. In this review, authors firstly summarized the resistance mechanisms in PDAC, and then emphasized the potential of repurposing antihypertensive drugs as an adjuvant therapeutic option in PDAC.
Overall, this review is informative and instructive, which will benefit the audiences in the field of cancer therapy and cardiovascular diseases. However, the underlying mechanism of the chemosensitizing effect of antihypertensive drugs still needs to be elucidated regarding the indirect anti-tumor or immune-regulation function. Importantly, it will be more instructive to supply potential research questions for clinicians and scientists. In addition, it's better to write out full names when firstly used such as COPD、ARB et al.
Author Response
Pancreatic ductal adenocarcinoma (PDAC) demonstrates resistance to commonly-used chemotherapy. Meanwhile, metabolic syndrome (MS) such as hypertension and obesity contributes to elevated risks of both cardiovascular diseases and PDAC. Thus, this review emphasizes that the repurposing of antihypertensive drugs may prove to be beneficial chemosensitizer for the treatment of unresectable PDAC patiens. In this review, authors firstly summarized the resistance mechanisms in PDAC, and then emphasized the potential of repurposing antihypertensive drugs as an adjuvant therapeutic option in PDAC.
Overall, this review is informative and instructive, which will benefit the audiences in the field of cancer therapy and cardiovascular diseases. However, the underlying mechanism of the chemosensitizing effect of antihypertensive drugs still needs to be elucidated regarding the indirect anti-tumor or immune-regulation function. Importantly, it will be more instructive to supply potential research questions for clinicians and scientists. In addition, it's better to write out full names when firstly used such as COPD、ARB et al.
Response:
We would like to express our gratitude to the reviewer for dedicating their time to a thorough analysis of our manuscript. With regard to the immunological mechanisms, the following sentences were included in the manuscript: “Due to lack of CD8+ T cells in the tumor microenvironment of PDAC, the strategies to inhibit the immune checkpoints fail to demonstrate efficacy in advanced PDAC patients [72]. Inflammatory markers amplify these effects, while myeloid-derived suppressor cells (MDSCs), regulatory T cells (Tregs), and M2 macrophages simultaneously suppress effector T cells, dendritic cells, and CD8+ cytotoxic T cells, hindering their ability to target and destroy tumor cells. “
Given the plethora of abbreviations utilized in the present manuscript, it was deemed preferable to forego an initial translation and instead provide a glossary at the conclusion of the manuscript. This approach enables the reader to readily access explanations of the discussed concepts at any point in the text.
We added the following hypotheses in the discussion section:” The multiplicity of potential signaling pathways implicated in chemoresistant pancreatic cancer may present a significant challenge in the development of an optimal therapeutic strategy. The MAPK/ERK pathway and potential mechanisms for targeting it may offer the greatest promise. The potential role of calcium ions in resistance may be a topic for discussion in clinical trials of CCBs and adjunctive chemotherapy. „
Reviewer 2 Report
Comments and Suggestions for Authors
This is a well written review on drug repositioning for pancreatic cancer.
1. Description in Introduction is too broad and can be shortened.
2. Effects of beta-blockers were reported. Please add data and discuss.
3. Table 1. needs to be reorgaized as human vs. non-human studies. Does original research mean non-human studies?
4. Sections 1 and 2 are different from the topic of this review. Please consider to delete or reorganize the sections.
Author Response
This is a well written review on drug repositioning for pancreatic cancer.
Firstly we would like to express our gratitude to the reviewer for dedicating their time to a thorough analysis of our manuscript.
- Description in Introduction is too broad and can be shortened.
Response: The fragments mentioned by the reviewer have been shortened.
- Effects of beta-blockers were reported. Please add data and discuss.
Response: The fragment pertaining to B-blockers was added: “ 3.3. Beta-blockers-BBs
Catecholamines regulate numerous cellular and tissue functions critical for the development and progression of many common cancers [120]. Adrenergic receptors are categorized into two main classes, α- and β-receptors, each with further subdivisions [121]. The β2-adrenergic receptor (β2AR), which primarily binds epinephrine and norepinephrine, is predominantly found in airway smooth muscle and cardiac tissue [121, 122]. However, its presence in other tissues, including the pancreas, has been identified more recently [123].
Beta-adrenergic receptor activation is believed to promote cell proliferation through PKA pathway activation. PKA signaling drives proliferation and inhibits apoptosis via downstream effectors such as the cAMP CREB, AP-1, and NF-κB [124]. Additionally, β-adrenergic activation increases the levels of MMP-2, MMP-9, and VEGF [125, 126, 127]. These molecules facilitate the degradation of extracellular matrix components and cell adhesion molecules, as well as the release and activation of growth factors [128,129, 130]. Thus, various complex, receptor-mediated signaling pathways are implicated in β-adrenergic receptor-driven process of angiogenesis, tumor proliferation, invasion, migration, metastasis in PDAC [ 125, 127, 131, 132, 133].
Epidemiological and retrospective data on β-blocker (BB) use in PDAC patients show mixed results. A Swedish population-based cohort study reported reduced cancer-specific mortality among PDAC patients using BBs, whereas retrospective analyses using UK and Clinical Practice Research Datalink databases provided inconsistent findings, potentially due to unadjusted cancer staging [134, 135].
A meta-analysis of 319,006 patients by Na et al. in 2018 demonstrated that BB use was associated with improved OS in pancreatic cancer, as well as in ovarian cancer (HR = 0.59, 95% CI: 0.36–0.96, P = 0.034) and melanoma (HR = 0.81, 95% CI: 0.67–0.97, P = 0.026). For pancreatic cancer specifically, the meta-analysis included two studies with 16,092 patients, showing an association between BB use and prolonged OS (HR = 0.85, 95% CI: 0.75–0.97, P = 0.014) [136].
Mechanistically, BBs inhibit key oncogenic pathways, including NF-κB, PI3K/Akt, and PKA. For instance, combining the β2-adrenoceptor blocker (ICI 118551) with gemcitabine has shown enhanced apoptosis in pancreatic cancer cells through regulation of pro-apoptotic (Bax) and anti-apoptotic (Bcl-2) molecules downstream of NF-κB. This combination provided superior tumor suppression compared to gemcitabine alone, highlighting a potential for synergy in chemotherapy [137].
With their established safety profile, affordability, and well-characterized pharmacology, BBs represent a promising adjunct to existing chemotherapy regimens in PDAC management. Future prospective studies are essential to explore their potential synergistic effects with chemotherapy, such as gemcitabine, and to better define the comparative benefits of selective and nonselective BBs in this setting.”
- Table 1. needs to be reorgaized as human vs. non-human studies. Does original research mean non-human studies?
Response: The table was revised according to the suggestions.
- Sections 1 and 2 are different from the topic of this review. Please consider to delete or reorganize the sections.
Response: It is acknowledged that sections 1 and 2 serve as an introduction to the topic, which may appear superfluous. However, from the perspective of a clinician who does not diagnose this condition on a daily basis, the introductory sections appear to be justified. Accordingly, we have elected to retain these sections in the manuscript.